# Socioeconomic inequalities in maternal healthcare utilization: An analysis of the interaction between wealth status and education, a population-based surveys in Tanzania

**Deogratius Bintabara**[1]*, **Ipyana Mwampagatwa**[2]

**1** Department of Community Medicine, The University of Dodoma, Dodoma, Tanzania, **2** Department of Obstetrics and Gynecology, The University of Dodoma, Dodoma, Tanzania

* bintabaradeo@gmail.com

## Abstract

Limited scientific, evidence has so far described the interactions between socioeconomic factors and the gap of inequalities in maternal healthcare utilization. This study assessed the interaction between wealth status and education to identify women with greater disadvantage. This analysis used secondary data from the three most recent rounds (2004, 2010, and 2016) of the Tanzania Demographic Health Survey (TDHS). Maternal healthcare utilization was assessed based on six services (outcomes) which are i) booking during the first trimester (bANC), ii) at least four antenatal visits (ANC4+), iii) adequate antenatal care (aANC), iv) facility-based delivery (FBD), v) skilled birth attendance (SBA), vi) cesarean section delivery (CSD). The concentration curve and the concentration index were used to measure socioeconomic inequality in maternal healthcare utilization outcomes. The interaction coefficients suggest that each unit increase in the wealth status is significantly associated with higher odds of utilizing all maternal healthcare services for women with primary and secondary or higher education compared to those with no education (booking during the first trimester [AOR = 1.30; 95% CI: 1.08–1.57], at least four antenatal visits [AOR = 1.16; 95% CI: 1.01–1.33], facility-based delivery [AOR = 1.29; 95% CI: 1.12–1.48], skilled birth attendance [AOR = 1.31; 95% CI: 1.15–1.49]). The highest wealth-related inequality in bANC (EI: 0.166), at least four antenatal visits (EI: 0.259), FBD (EI: 0.323) and skilled birth attendance (EI: 0.328) ($P < 0.05$) was observed among women with primary and secondary or higher education. These findings provide strong evidence that there is an interaction effect between education attainment and wealth status in socioeconomic inequalities of maternal health services utilization. Therefore, any approach which will address both women's education and wealth status might be the first step to reducing socioeconomic inequalities in maternal health services utilization in Tanzania.

**Data Availability Statement:** The datasets generated during the current study are available in

the Demographic and Health Survey Program repository: http://dhsprogram.com/data/available-datasets.cfm and are accessible after registration on the website. The authors did not have special access privileges.

**Funding:** The authors received no specific funding for this work.

**Competing interests:** The authors have declared that no competing interests exist.

## Introduction

Over the past three decades, the world has witnessed a substantial decline in maternal mortality ratio (MMR). However, this achievement is not much experienced in developing countries and Sub-Saharan Africa (SSA) which accounts for about 99% and 68% of all global maternal deaths respectively [1, 2]. In 2017, SSA reported 533 maternal deaths per 100,000 live births which is the highest ratio compared to any other region [2]. Gender, education and literacy, and economic factors have been highlighted as probable social determinants of maternal mortality within SSA [3]. These factors have been associated with difficulties in access to maternal health care services during pregnancy and delivery which resulted in high MMR in this region [4, 5]. Existing studies indicate the persistence of socioeconomic inequalities as the major contributor to difficulties in access to maternal health care services [6]. This implies women from lower socioeconomic status are less utilizing maternal healthcare services for pregnancy and delivery. Simply most of these women especially in SSA have insufficient access to financial resources which jeopardizes their health-seeking behavior [3].

Tanzania still has a low proportion of pregnant women who effectively utilize antenatal care (ANC) and skilled delivery. Recently report showed that only 24% of pregnant women booked for ANC in the first trimester (bANC), 51% attends at least four ANC visits (ANC4+), 63% accessed facility-based delivery (FBD) and 64% received skilled birth attendance during delivery (SBA) [7]. Evidence from previous studies documented the close linkage between low coverage and inequality in the utilization of maternal healthcare services in favor of wealthier groups [6, 8]. For instance, one study conducted in Tanzania reported the higher odds of utilizing maternal healthcare services for wealthier women compared to poorer women [9].

Besides, the effect of wealth status other studies highlighted educational factors as another contributor to inequalities in the utilization of maternal healthcare services in favor of educated ones [10, 11]. In a study conducted in the Mwanza region in western Tanzania, It was documented that educated women were more likely to utilize ANC and FBD [12]. Therefore, it is unclear whether interactions of these two factors expose women to high inequalities in utilization of maternal healthcare services as most previous studies assessed and discussed inequalities without considering the effect of their interactions. The limited evidence regarding whether interactions between education and wealth status increase the gap of inequalities in maternal healthcare utilization suggested the need for this analysis to identify groups of women at a greater disadvantage.

## Methods

### Data source

This study was performed using secondary data from the three most recent rounds (2004, 2010, and 2016) of the Tanzania Demographic Health Survey (TDHS). The TDHS is the part of global Demographic Health Survey (DHS) program.

### Sampling technique

The TDHS employed two-stage cluster sampling methods. In the first stage, the clusters as primary sampling units were selected [13]. In the second stage, a total of 22 households were systematically selected from each cluster, yielding a representative probability sample of 10,312, 10,300, and 13,376 households for 2004, 2010, and 2016 TDHSs. The following women aged 15 to 49 years were interviewed from the surveyed household: 10,139 women in 2004; 10,329 women in 2010; and 13,266 women in 2016 (average response rate of 97%). In this analysis, a total sample of 14,445 and 22,892 ever-married women aged 15–49 years (from 2004 to 2016)

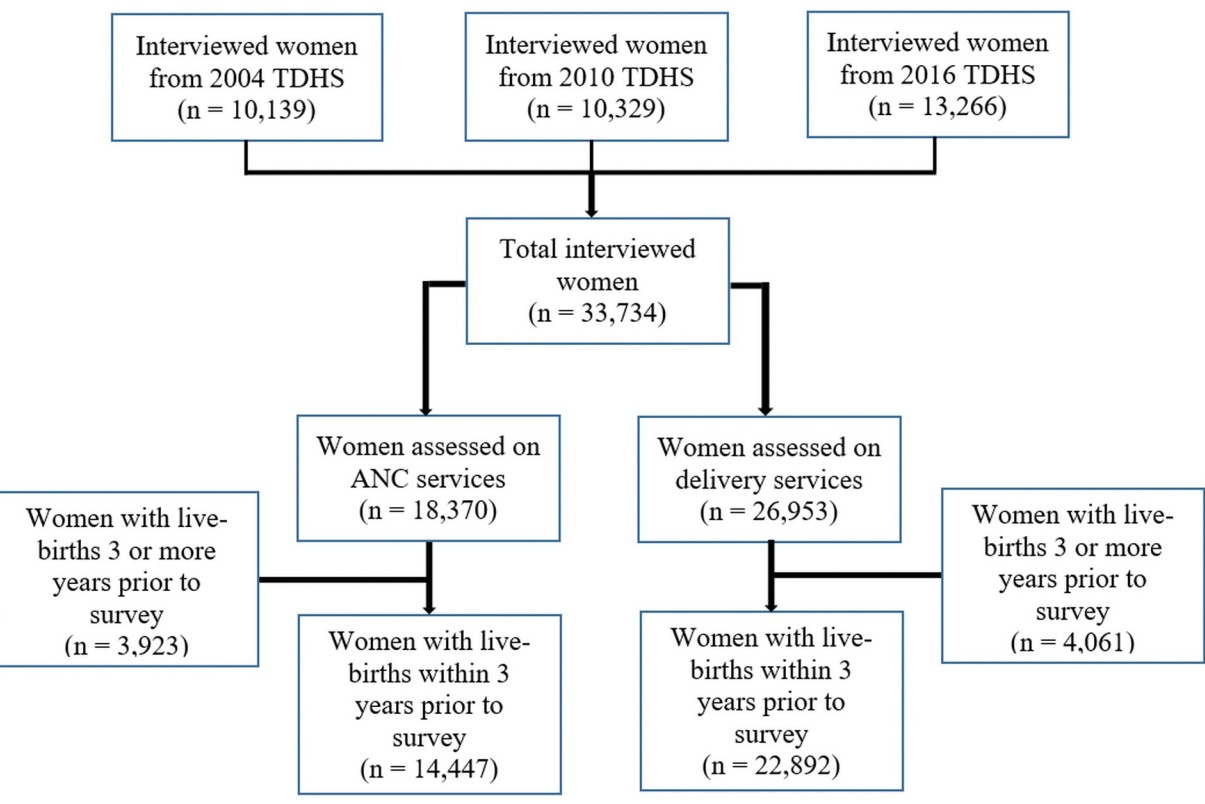

**Fig 1. Selection of study respondents included in this analysis.**

with at least one live birth in the three years before the survey were included for the assessment of ANC and delivery care services utilization respectively (Fig 1).

## Measurement of variables

**Outcome variables.** The six outcome variables grouped into two major categories of maternal healthcare utilization were analyzed in this study. The first category is related to ANC utilization which included three outcomes (bANC, ANC4+, and aANC). (i) bANC was considered "Yes" if women booked ANC in the first trimester otherwise consider "No;" (ii) ANC4+ was considered "Yes" if women attended at least four visits, otherwise considered "No;" and (iii) aANC was considered "Yes" if a woman reported receiving all of the following; (i) iron/folate tablets or syrup, (ii) antimalarial drugs, (iii) blood pressure measured (iv) urine sample taken (v) blood sample taken (vi) intestinal parasite drugs, otherwise considered "No."

The second category related to the utilization of delivery care services which also included three outcomes (FBD, SBA, and CSD). (i) FBD which was considered "Yes" when a woman delivered at any of the following public or private facilities; clinic, dispensary, health center, or hospital, otherwise considered "No." (ii) SBA which was considered "Yes" if woman assisted by the following trained health personnel; assistant nurse, nurse/ midwife, clinical assistant, clinical officer, assistant medical officer, medical officer, and medical specialist, otherwise considered "No." CSD which was considered "Yes" if women delivered by cesarean section delivery otherwise considered "No."

**Independent variables.** The primary independent variables were (i) household wealth index and (ii) education level. The wealth index in TDHS was calculated based on household

assets and housing characteristics. The principal component analysis was used to assign a score to each household based on selected household assets. From the continuous index score the households were ranked by dividing the sample population distribution into five equal categories ("poorest," "poorer," "middle," "richer" and "richest"), each including 20% of the population. This approach has been explained in detail elsewhere [14]. The education level was categorized as "none" for women who had not received any kind of formal education, "primary" for women who completed primary education level, "secondary" for women who completed secondary education level, and "highest" for women who completed college and all university level.

The following were adjusting factors used in this study; women's age was grouped into "15–19," "20–24," "25–29," "30–34," "35–39," "40–44," and "45–49." Place of residence was categorized as "urban" for households located in cities, municipalities, and town councils gazetted under the Local Government Act, 1982 [15], and "rural" for households that were located outside the urban areas. Marital status was categorized as "no spouse" for single women, divorced, separated, or widowed and "living with spouse" for women who were married or living with their partner during the interview. Employment was categorized as "employed" for women who reported to be employed and paid in salary in terms of cash and "not employed" for those who did not have any kind of job and were paid in terms of cash. These variables were selected based on previous studies that assessed the correlates of maternal healthcare utilization in Tanzania [16, 17].

## Statistical analysis

The analysis of the current study was based on three methods; descriptive analysis, multivariable logistic regression, and assessment of wealth-related inequality by education level. The multivariable logistic regression analyses were performed to assess the relationship between the household wealth index and each of all six outcomes of maternal healthcare utilization. The interactions between household wealth index (as a continuous variable) and education level regarding the six outcomes were considered to assess if the observed relationships vary across the education levels. A similar approach has been used previously in other studies [8, 18]. Furthermore, the following two approaches were used to evaluate socioeconomic inequalities in all six outcomes of maternal healthcare utilization. The methods were the construction of Concentration Curves (*CC*) and the computation of Concentration Indices (*Ci*).

**Concentration curves.** The *CC* was used to evaluate the patterns of socioeconomic inequalities for each outcome of maternal healthcare utilization. The *CC* plot the cumulative percentage of the outcome variable (y-axis) against the cumulative percentage of the population ranked by household socioeconomic status (using a raw score of wealth index), beginning with the poorest, and ending with the richest (x-axis) [19, 20]. In other words, they plot shares of the outcome variable against quintiles of the household wealth index.

**Concentration indices.** The computation of *Ci* has an additional advantage compared to *CC* for quantifying the degree of socioeconomic-related inequality in healthcare variables such as the utilization of maternal healthcare services. This *Ci* is defined as "twice the area between *CC* and the line of equality." It takes the values bounded between -1 and +1. If the health variable is "good" such as the utilization of maternal healthcare, an index with a positive value indicates utilization is more among the rich [19, 20]. Mathematically, the *Ci* can be calculated by using the following formula (1).

$$Ci = \frac{2}{\mu} cov(y, r) \tag{1}$$

Because all six outcomes of maternal healthcare utilization were binary, the bounds of the

*Ci* may not lie between −1 and +1. The two techniques which produce Wagstaff and Erreygers *Ci* are frequently used to account for the aforementioned limitation [21, 22]. Therefore, this study used the Erreygers *Ci* which satisfies four properties of the rank-dependent measure of socioeconomic inequality [8, 23].

### Ethical considerations

This study was based on an analysis of existing public domain survey datasets that are freely available online with all identifier information removed. The surveys were approved by the National Institute of Medical Research Ethics Committee in Tanzania. Therefore, ethics approval for the current analysis was automatically deemed unnecessary. Informed consent was requested and obtained from the participants before the TDHS interviews. For participants under 16 years old the consent was obtained from their parent or guardian. All participants who accepted to participate in the surveys were provided a signed written informed consent.

### Results

Table 1 shows the distribution of women by maternal healthcare utilization indicators. It demonstrates that the proportion of utilization of these six maternal healthcare (bANC, ANC4+, aANC, FBD, SBA, and CSD) were higher among women with secondary or higher education (29%, 67%, 50%, 87%, 88%, and 15%) compared to those without formal education (13%, 42%, 14%, 35%, 35%, 1%) respectively. Examining the wealth status we found that the richest women have a higher proportion in all six indicators of maternal healthcare utilization compared poorest women. Similarly, maternal healthcare utilization (all six indicators) was observed to be higher among women in urban compared to rural areas.

Table 2 shows the richest to poorest ratios of maternal healthcare utilization indicators according to women's education level. The results indicate a higher variation in maternal healthcare utilization between the richest and poorest women across all levels of education. This revealed that richer women utilized more maternal healthcare services compared to their poorer counterparts. For example, the richest to the poorest ratio for bANC increased from 0.92 (no education) to 1.74 (primary) and 1.46 (secondary/higher). This indicates education has a significant effect on increasing the utilization of maternal healthcare.

Table 3 presents the adjusted odds ratios (OR) with a 95% confidence interval (CI) from multivariable logistic regression models. All models were adjusted for woman's age, marital status, place of residence, year of survey, and employment status. Also, the models included the interaction term between wealth status (as continuous) and woman's education level. The significant result of the interaction term indicates the strong evidence that women's education has effect modification in the association between wealth status and maternal healthcare utilization. The interaction coefficients suggest that each unit increase in the wealth status is significantly associated with higher odds of utilizing all maternal healthcare services (for women with primary and secondary or higher education compared to those with no education) except for receiving aANC and CSD. This effect seems to be much higher for women with secondary or higher education. Overall the findings indicate that the wealth-related gradient in maternal healthcare utilization varies significantly across the level of education in Tanzania.

Fig 2 displays adjusted predicted probabilities of maternal healthcare utilization by the interaction of wealth status and education obtained from regression models presented in Table 3. The predicted probabilities for maternal healthcare utilization outcomes increase sharply from poorest to richest for women with primary and secondary or higher education compared to those with no education. This means the wealth gradient in maternal healthcare

**Table 1. Descriptive statistics of women included in the analysis by maternal healthcare utilization indicators.**

| | Booking ANC during 1st trimester | At least four ANC visits | Receiving adequate ANC | Facility-based delivery | Skilled birth attendance | Cesarean section delivery |
|---|---|---|---|---|---|---|
| **Woman age** | | | | | | |
| 15–19 | 14.79 | 49.17 | 24.39 | 60.69 | 61.22 | 4.50 |
| 20–24 | 17.84 | 50.63 | 25.08 | 55.81 | 55.73 | 3.81 |
| 25–29 | 18.82 | 50.57 | 26.65 | 52.27 | 52.48 | 4.41 |
| 30–34 | 17.21 | 50.59 | 26.47 | 50.71 | 50.74 | 4.74 |
| 35–39 | 15.31 | 46.27 | 23.18 | 45.47 | 46.07 | 4.00 |
| 40–44 | 15.69 | 47.18 | 23.83 | 44.68 | 45.04 | 2.78 |
| 45–49 | 15.36 | 43.93 | 16.57 | 35.76 | 37.06 | 2.79 |
| **Woman education** | | | | | | |
| None | 13.36 | 42.44 | 13.65 | 34.83 | 34.71 | 1.40 |
| Primary | 16.69 | 49.50 | 25.50 | 53.91 | 54.10 | 3.76 |
| Secondary and higher | 29.31 | 66.89 | 50.07 | 87.31 | 88.49 | 15.44 |
| **Wealth status** | | | | | | |
| Poorest | 13.54 | 39.61 | 16.54 | 37.91 | 38.15 | 1.79 |
| Poor | 14.23 | 44.85 | 15.78 | 39.45 | 39.32 | 1.89 |
| Middle | 14.60 | 46.35 | 18.35 | 43.76 | 43.76 | 2.95 |
| Rich | 16.88 | 52.45 | 26.89 | 55.49 | 56.02 | 3.40 |
| Richest | 26.69 | 64.79 | 48.55 | 88.06 | 88.40 | 10.73 |
| **Marital status** | | | | | | |
| No spouse | 18.77 | 50.12 | 29.26 | 60.88 | 61.52 | 5.25 |
| Living with spouse | 16.88 | 49.51 | 24.43 | 50.50 | 50.62 | 3.97 |
| **Employment** | | | | | | |
| Not employed | 15.53 | 48.17 | 20.23 | 46.30 | 46.33 | 2.87 |
| Employed | 20.34 | 52.33 | 34.68 | 63.64 | 64.17 | 6.79 |
| **Residence** | | | | | | |
| Rural | 15.11 | 45.84 | 18.44 | 43.79 | 43.95 | 2.52 |
| Urban | 23.92 | 61.81 | 47.16 | 82.83 | 83.16 | 10.31 |
| **Year of survey** | | | | | | |
| 2004 | 13.56 | 59.24 | 15.96 | 45.45 | 44.53 | 2.91 |
| 2010 | 13.85 | 39.77 | 21.91 | 47.92 | 48.23 | 3.96 |
| 2016 | 22.94 | 49.21 | 35.75 | 61.15 | 62.24 | 5.43 |
| Total | 17.19 | 49.61 | 25.22 | 51.96 | 52.15 | 4.15 |

**Table 2. Richest to poorest ratios of maternal healthcare utilization by education level.**

| Outcome variable | Education level | | | Total |
|---|---|---|---|---|
| | None | Primary | Secondary/higher | |
| **Booking ANC during 1st trimester** | 0.92 | 1.74 | 1.46 | 1.97 |
| **At least four ANC visits** | 1,41 | 1.52 | 1.44 | 1.64 |
| **Adequate ANC** | 2.92 | 2.41 | 1,44 | 2.94 |
| **Facility-based delivery** | 2.19 | 2.12 | 1.36 | 2.32 |
| **Skilled birth attendance** | 2.15 | 2.08 | 1.30 | 2.32 |
| **Cesarean section delivery** | 0.69 | 4.71 | 3.96 | 6.62 |

**Table 3. Multivariate logistic regression models with an interaction term between wealth index and education.**

| | Booking ANC during 1st trimester | | At least four ANC visits | | Receiving adequate ANC | | Facility-based delivery | | Skilled birth attendance | | Cesarean section delivery | |
|---|---|---|---|---|---|---|---|---|---|---|---|---|
| | OR | 95% CI | OR | 95% CI | OR | 95% CI | OR | 95% CI | OR | 95% CI | OR | 95% CI |
| **Wealth** | 1.05** | 0.95–1.16 | 1.09* | 1.01–1.19 | 1.26** | 1.14–1.41 | 1.17** | 1.07–1.27 | 1.17** | 1.08–1.27 | 1.00 | 0.78–1.28 |
| **Education** | | | | | | | | | | | | |
| None (reference) | | | | | | | | | | | | |
| Primary | 0.81 | 0.59–1.11 | 1.02 | 0.81–1.29 | 1.58** | 1.15–2.16 | 1.10 | 0.85–1.41 | 1.07 | 0.85–1.36 | 1.08 | 0.53–2.16 |
| Secondary and higher | 0.59 | 0.28–1.26 | 1.13 | 0.68–1.89 | 2.60** | 1.42–4.74 | 1.80* | 1.06–3.05 | 1.90* | 1.13–3.17 | 2.80 | 0.77–10.17 |
| **Education x wealth** | | | | | | | | | | | | |
| None x wealth (reference) | | | | | | | | | | | | |
| Primary x wealth | 1.13* | 1.01–1.26 | 1.06 | 0.96–1.16 | 0.99 | 0.88–1.10 | 1.16* | 1.06–1.26 | 1.17* | 1.08–1.28 | 1.25 | 0.97–1.62 |
| Secondary/higher x wealth | 1.30* | 1.08–1.57 | 1.16* | 1.01–1.33 | 0.95 | 0.81–1.10 | 1.29** | 1.12–1.48 | 1.31* | 1.15–1.49 | 1.24 | 0.88–1.74 |

Note:

* p < 0.05

** p < 0.01

CI = Confidence Interval

Wealth has been used as continuous variable.

All models are adjusted for age, marital status, place of residence, year of survey, and employment status.

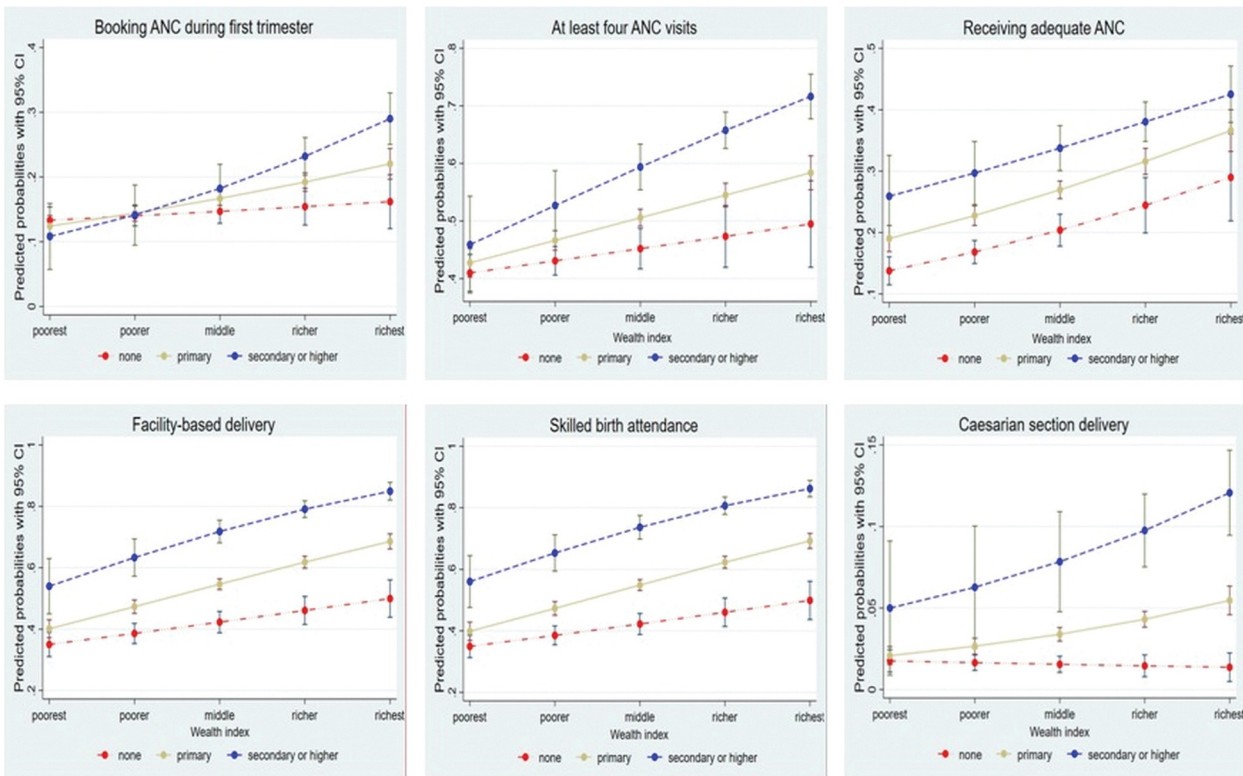

**Fig 2. Adjusted predicted probabilities of maternal healthcare utilization outcomes by education level and wealth index.**

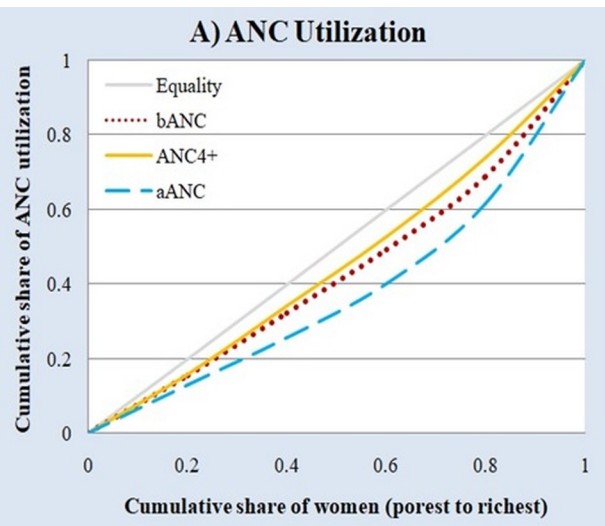
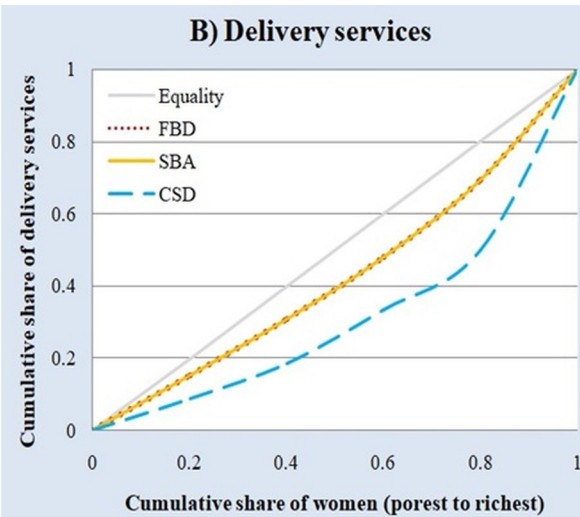

**Fig 3. Concentration curves of maternal healthcare utilization outcomes.** (A) Concentration curves for the three ANC utilization outcomes. (B) Concentration curves for the three delivery services utilization outcomes.

utilization is steeper due to the effect of educational attainment. The flatter predicted probabilities curves for women with no education indicate a less steep wealth gradient in maternal healthcare utilization. This means, there was a low difference in probabilities of maternal healthcare utilization between poor and rich with no education.

The CCs were used to assess the presence of inequalities in maternal healthcare utilization. Fig 3A and 3B displays the CCs for ANC and delivery utilization services respectively. For all six outcomes, the CCs are below the line of inequality suggesting pro-rich inequalities in maternal healthcare utilization. For example, for ANC utilization, the poorest 40% of women accounted for 34% of ANC4+, but it was only 25% for receiving aANC.

The normalized CI called Erreygers's index (EI) was used to show the magnitude and comparison of inequalities of the outcome variables across education levels. As shown in Figs 4 and 5, all EI were positive and significantly different from zero. This suggests the utilization of both ANC and delivery services across all education levels was in favor of women from wealthier households. For example, in Fig 4, the highest wealth-related inequality in bANC (EI: 0.166) and ANC4+ (EI: 0.259) was observed among women with secondary or higher education.

## Discussion

This study aimed to examine whether interactions between maternal education and household wealth status influence inequalities in maternal healthcare utilization in Tanzania. The findings revealed that for the past two decades, there was a substantial pro-rich inequality in all six maternal healthcare utilization outcomes. This suggests that richer women were generally advantaged to utilize essential maternal healthcare services probably due to fewer barriers to accessing healthcare compared to women from the poorest households [24]. Furthermore, the higher concentration of all six maternal healthcare utilization outcomes among the richer women could be explained by the fact that wealthier households can afford to purchase goods and services including paying the cost of maternal healthcare services [25]. The existing inequalities in utilizing maternal healthcare services indicate the uneven distribution of initiatives, programs, and their effects as major determinants [26]. For example, for the past two

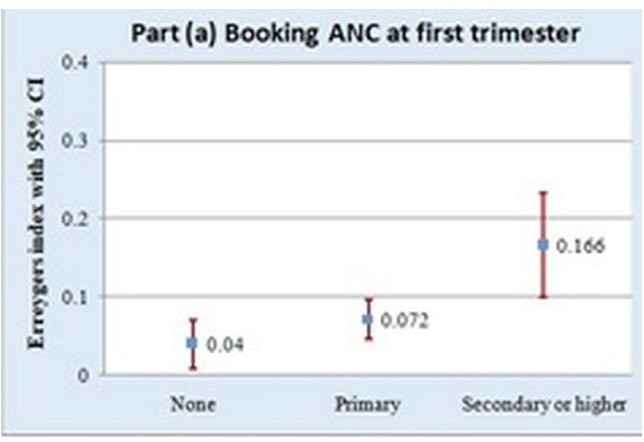

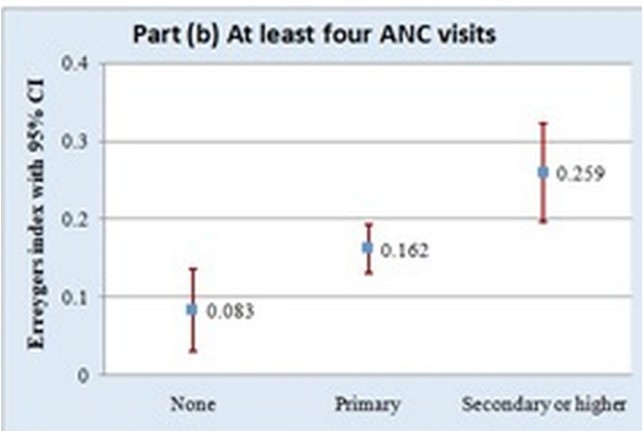

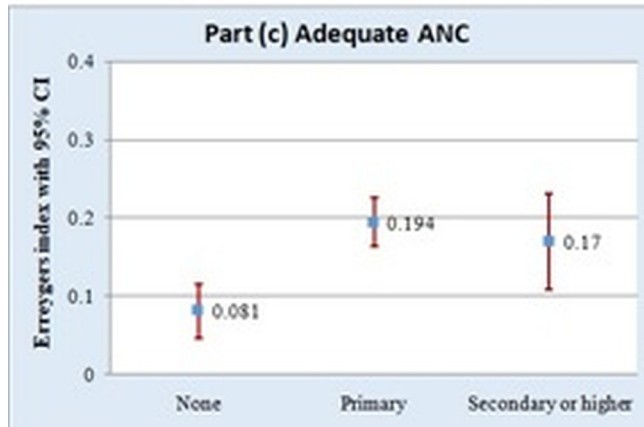

**Fig 4. Inequality in ANC utilization outcomes according to education levels.**

decades, much effort has been made including increased coverage of ANC, number of health facilities, and training staff to provide skilled delivery services but all seem to benefit women from wealthier households than poor ones. Also, exemption and waiver directives for the cost of maternal health services seem not the right approach to eliminate inequalities. This might be due to a lack of legal weight for its effective implementation which exposes women to maternal healthcare-related costs [27, 28]. Putting any cost for maternal healthcare services can be easily affordable to richer women hence resulting in observed inequalities. Therefore,

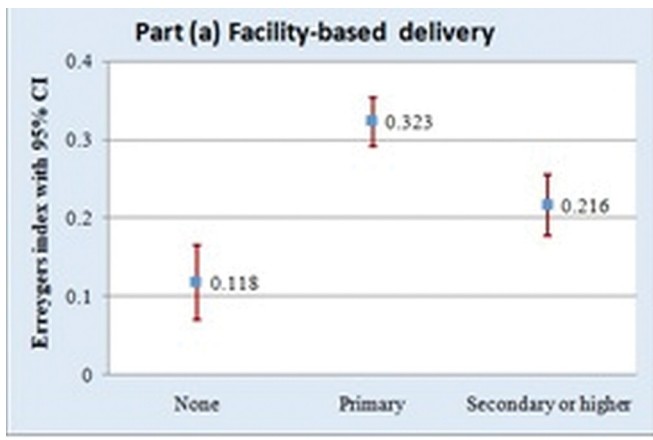

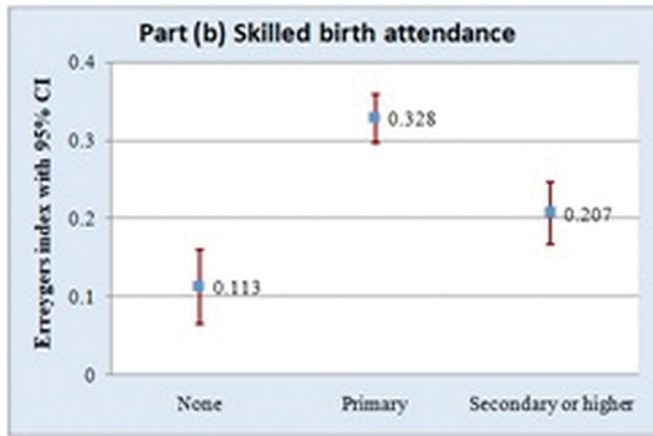

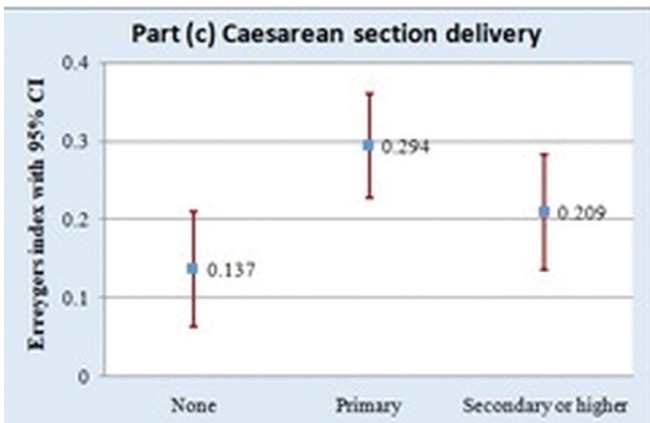

**Fig 5. Inequality in delivery service utilization outcomes according to education levels.**

efforts to the reduction of wealth-related inequalities are greatly needed to achieve universal maternal healthcare coverage in Tanzania.

Across all maternal healthcare services utilization, our findings indicate that inequalities are higher (pro-rich) for receiving delivery services compared to ANC services. This might be because most ANC services are offered for free in both public and private facilities. Furthermore, it might be due to the difference in coverage of these two groups of maternal healthcare services in Tanzania. The current estimates show that there is high coverage of ANC (>95%)

compared to delivery services (64%) [7]. For ANC utilization the high pro-rich inequalities are observed in receiving aANC while for delivery services is accessing CSD. This might be because the two services require skilled health professionals who sometimes special consultations for cost are needed and these might be affordable for richer women compared to poorer women. Similar findings have been reported in the study conducted in Bangladesh [8].

Evidence from previous studies showed that women's education is significantly associated with an increase in maternal healthcare utilization in developing countries [29, 30]. This indicates that exposure to education for women is crucial to enhancing the utilization of maternal healthcare services. However, there is a piece of limited information on whether exposure to education has the same effect on different socioeconomic statuses. Therefore, our findings addressed this gap and demonstrate that the effect of increased women's education on the use of maternal healthcare services appears to be stronger for women from wealthier households and weaker for women from poorer households. This has been demonstrated by the steeper wealth-related gradient in maternal healthcare utilization with education level attainment in all services. Therefore, encouraging education only will have less effect on poorer women compared to richer women in the utilization of maternal healthcare services. This means any education intervention has to consider the variation of socioeconomic inequalities among women in Tanzania. Therefore, to reduce these inequalities, a multi-sectorial approach is needed so that women can be empowered socioeconomically (both education and financial) for the country to achieve universal maternal healthcare coverage on time. This can be possible by considering integrated policy interventions which include fiscal policies, government spending, social protection, labor market, employment policies, and others.

Socioeconomic inequalities in maternal healthcare services utilization observed in this study considerably varied across educational levels of Tanzania. The observed high pro-rich inequalities for women with primary and secondary or higher education compared to those without education suggests that being rich alone does not guarantee the woman to receive, seek or access better healthcare. However, it places her in a good position to overcome one of the biggest barriers (money) to accessing maternal healthcare services in Tanzania [24]. On other hand, low pro-rich inequalities for women without education, signify that if all women would have no education, there might be low or insignificant variation in maternal healthcare services utilization across all levels of wealth status. The findings from this study underscore the importance of women's education and wealthier status in maternal healthcare services utilization [31–33]. It is well documented that empowering and providing opportunities for scholarship in women's education would increase women's health literacy, knowledge about maternal and child health issues, a sense of personal control, and access to healthcare facilities. Therefore, the integrated approach interventions focusing to address both education and wealth-related factors would result in many effects on reducing/eliminating the gap of inequalities in maternal healthcare utilization. This might be an initial strategy toward achieving universal maternal healthcare coverage in Tanzania.

To the best of our knowledge, our study is the first of its kind to examine socioeconomic inequalities in maternal healthcare utilization in Tanzania by considering the interaction effect between education and wealth status. The use of data from three consecutive rounds of nationally representative surveys with a high response rate and robust sampling techniques, assures our findings provide accurate estimates.

However, this study had some limitations, the use of data from the cross-sectional surveys limits the causal interpretation of the relationship between outcome variables and education and wealth status. Therefore, the results should be interpreted with caution. This study is not beyond limitations. Reporting bias, because data on maternal healthcare utilization were based on self-reported. This might have affected the reliability of the estimates. Also, a risk of recall

bias might have been introduced as a result of the long recall period, however, this has been minimized by using data from the women with most recent live births. Furthermore, some of the findings can be a basis for further studies. For instance, according to Multivariate logistic regression models with an interaction term between wealth index and education there is no association between CSD and wealth and education. However, descriptive statistics indicate the rate of CSD is higher among the rich and richest quintile and among women with the secondary and higher-level education attainment. Therefore, in this study we discussed the interaction effect of wealth index and education by using adjusted predicted probabilities which showed clearer interpretation than multivariate analysis.

## Conclusions

The current analysis provides strong evidence that there is an interaction effect between education attainment and wealth status in socioeconomic inequalities of maternal health services utilization. The study provides an empirical basis for policies, strategies, and interventions focusing on both empowering women's education and efforts to improve their wealth status. Therefore, any approach which will address both women's education and wealth status might be the first step to reducing socioeconomic inequalities in maternal health services utilization in Tanzania.

## Acknowledgments

We would like to acknowledge ICF International, Rockville, Maryland, USA, through DHS program for giving us permission to access the 2004, 2010, and 2016 Tanzania DHS datasets.

## Author Contributions

**Conceptualization:** Deogratius Bintabara.

**Data curation:** Deogratius Bintabara.

**Formal analysis:** Deogratius Bintabara.

**Methodology:** Deogratius Bintabara.

**Project administration:** Deogratius Bintabara, Ipyana Mwampagatwa.

**Resources:** Ipyana Mwampagatwa.

**Software:** Deogratius Bintabara.

**Supervision:** Ipyana Mwampagatwa.

**Validation:** Ipyana Mwampagatwa.

**Visualization:** Deogratius Bintabara, Ipyana Mwampagatwa.

**Writing – original draft:** Deogratius Bintabara, Ipyana Mwampagatwa.

**Writing – review & editing:** Deogratius Bintabara, Ipyana Mwampagatwa.

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
