## [Decision Letter · Decision Letter 0]

4 Apr 2023

PGPH-D-23-00005

Socioeconomic inequalities in maternal healthcare utilization: an analysis of interaction between wealth status and education, a population-based surveys in Tanzania

Dear Dr. Bintabara,

Thank you for submitting your manuscript to PLOS Global Public Health. After careful consideration, we feel that it has merit but does not fully meet PLOS Global Public Health’s publication criteria as it currently stands. Therefore, we invite you to submit a revised version of the manuscript that addresses the points raised during the review process.

We look forward to receiving your revised manuscript.

Kind regards,

Samiratou Ouédraogo, DPharm, MPH, Ph.D.

Academic Editor

Journal Requirements:

Additional Editor Comments (if provided):

Reviewers' comments:

Reviewer's Responses to Questions

**Comments to the Author**

1. Does this manuscript meet PLOS Global Public Health’s publication criteria? Is the manuscript technically sound, and do the data support the conclusions? The manuscript must describe methodologically and ethically rigorous research with conclusions that are appropriately drawn based on the data presented.

Reviewer #1: Yes

Reviewer #2: Yes

2. Has the statistical analysis been performed appropriately and rigorously?

Reviewer #1: Yes

Reviewer #2: I don't know

3. Have the authors made all data underlying the findings in their manuscript fully available (please refer to the Data Availability Statement at the start of the manuscript PDF file)?

Reviewer #1: No

Reviewer #2: Yes

4. Is the manuscript presented in an intelligible fashion and written in standard English?

Reviewer #1: Yes

Reviewer #2: Yes

5. Review Comments to the Author

Reviewer #1: Abstract

In the background section of abstract authors have mentioned that there is still huge inequalities in

accessing maternal health service throughout pregnancy and postpartum periods. However, I this statement suggested as in this analysis the authors have measured the equity gap in utilization of service rather than access of service so suggest highlighting disparity in utilization rather than access in background section to make it more aligned with study purpose.

Similarly, in the method section of abstract, there is a need to describe method a bit more in the method section of abstract specially describe which data has been analyzed (DHS) as part of data collection method, describe characteristic of sample population, size including Period of study as this analysis has taken 3 set of TDHS.

Overall, the secondary analysis was conducted correctly with interesting findings. However, some of the findings can be a basis for further studies. For instance, according to Multivariate logistic regression models with an interaction term between wealth index and education there is no association between CSD and wealth and education however if we look at Descriptive statistics of women included in the analysis by maternal healthcare utilization indicators the rate of CSD in higher among the rich and richest quintile and among women with the secondary and higher-level education attainment. In general, CSD is higher in educated and wealthier as they often reach-out to the private sector also delay marriage, advance maternal age as educated often prefer delay marriage and pregnancy due to career related opportunity. Study from SSA also suggest https://www.ncbi.nlm.nih.gov/pmc/articles/PMC8975863/. CS and its association with educational attainment and wealth index however, this is not in the case of this analysis which need to be highlight in the discussion section and suggest for further study.

Reviewer #2: Iverall manuscript looks well written and is of importance to add knowledge on the coorelation of maternal healthcare with socioeconomic status and education of women, I have suggested some minor changes and I think this manuscript needs to be checked by a statistician to evaluate the findings of the study and validity of the methods used,

6. PLOS authors have the option to publish the peer review history of their article (what does this mean?). If published, this will include your full peer review and any attached files.

**Do you want your identity to be public for this peer review?** For information about this choice, including consent withdrawal, please see our Privacy Policy.

Reviewer #1: **Yes: **Adweeti Nepal

Reviewer #2: No

---

## [Decision Letter · Decision Letter 1]

9 May 2023

Socioeconomic inequalities in maternal healthcare utilization: an analysis of the interaction between wealth status and education, a population-based surveys in Tanzania

PGPH-D-23-00005R1

Dear Dr Bintabara,

We are pleased to inform you that your manuscript 'Socioeconomic inequalities in maternal healthcare utilization: an analysis of the interaction between wealth status and education, a population-based surveys in Tanzania' has been provisionally accepted for publication in PLOS Global Public Health.

Best regards,

Julia Robinson

Executive Editor

Reviewer Comments (if any, and for reference):

Reviewer's Responses to Questions

**Comments to the Author**

1. If the authors have adequately addressed your comments raised in a previous round of review and you feel that this manuscript is now acceptable for publication, you may indicate that here to bypass the “Comments to the Author” section, enter your conflict of interest statement in the “Confidential to Editor” section, and submit your "Accept" recommendation.

Reviewer #1: All comments have been addressed

2. Does this manuscript meet PLOS Global Public Health’s publication criteria? Is the manuscript technically sound, and do the data support the conclusions? The manuscript must describe methodologically and ethically rigorous research with conclusions that are appropriately drawn based on the data presented.

Reviewer #1: Yes

3. Has the statistical analysis been performed appropriately and rigorously?

Reviewer #1: I don't know

4. Have the authors made all data underlying the findings in their manuscript fully available (please refer to the Data Availability Statement at the start of the manuscript PDF file)?

Reviewer #1: No

5. Is the manuscript presented in an intelligible fashion and written in standard English?

Reviewer #1: Yes

6. Review Comments to the Author

Reviewer #1: The author has addressed all the recommendations in revised manuscript. The result and recommendation of analysis is very important for the policy makers while designing socio-economic approaches to address barrier to access and utilize maternal health services. However, I request editorial team to consider statistical review to validate statistical analysis and interpretation.

7. PLOS authors have the option to publish the peer review history of their article (what does this mean?). If published, this will include your full peer review and any attached files.

**Do you want your identity to be public for this peer review?** For information about this choice, including consent withdrawal, please see our Privacy Policy.

Reviewer #1: **Yes: **Adweeti Nepal
